# Distinct Molecular Profiles Underpin Mild-To-Moderate Equine Asthma Cytological Profiles

**DOI:** 10.3390/cells13221926

**Published:** 2024-11-20

**Authors:** Anna E. Karagianni, Eric A. Richard, Marie-Pierre Toquet, Erika S. Hue, Anne Courouce-Malblanc, Bruce McGorum, Dominic Kurian, Judit Aguilar, Stella Mazeri, Thomas M. Wishart, Robert Scott Pirie

**Affiliations:** 1School of Veterinary Medicine, Faculty of Health and Medical Sciences, VSM Building, University of Surrey, Daphne Jackson Road, Guildford, Surrey GU2 7AL, UK; 2The Roslin Institute and Royal (Dick) School of Veterinary Studies, University of Edinburgh, Easter Bush, Midlothian EH25 9PS, UK; bruce.mcgorum@ed.ac.uk (B.M.); dominic.kurian@roslin.ed.ac.uk (D.K.); jaguilar@anapharmbioanalytics.com (J.A.); smazeri@ed.ac.uk (S.M.); t.m.wishart@ed.ac.uk (T.M.W.); 3LABÉO, 14280 Saint-Contest, France; eric.richard@laboratoire-labeo.fr (E.A.R.); marie-pierre.toquet@laboratoire-labeo.fr (M.-P.T.); erika.hue@laboratoire-labeo.fr (E.S.H.); 4Université de Caen Normandie, BIOTARGEN UR7450, Normandie Univ, 14000 Caen, France; 5Centre International de Santé du Cheval d’Oniris (CISCO), Route de Gachet, 44307 Nantes, France; acourouce@orange.fr; 6Centre for Systems Health and Integrated Metabolic Research, Nottingham Trent University, Nottingham NG1 4GG, UK

**Keywords:** equine, bronchoalveolar lavage, transcriptomic, proteomic, airway immunity, asthma

## Abstract

A state-of-the-art multi-omics approach was applied to improve our understanding of the aetio-pathogenesis of a highly prevalent, performance-limiting disorder of racehorses: mild-to-moderate equine asthma (MMEA). This is a prerequisite to improving prophylactic, management, and therapeutic options for this condition. Although a number of risk factors have been identified, options for intervention are limited. This study applied a multi-omic approach to reveal key inflammatory pathways involved in inflammatory cell recruitment to the lower airways and highlight distinct MMEA inflammatory profiles. We compared bronchoalveolar lavage fluid (BALF) cell gene and protein expression data from horses with non-inflammatory BALF cytology with those isolated from horses with neutrophilic, mastocytic, mixed neutrophilic/mastocytic, and eosinophilic/mastocytic inflammation. The analyses on transcriptomic/proteomic data derived from BALF from horses with neutrophilic cytology showed enrichment in classical inflammatory pathways, and horses with mastocytic inflammation showed enrichment in pathways involved in hypersensitivity reactions related to nonclassical inflammation potentially mimicking a Th2-immune response. The mixed eosinophilic/mastocytic group also presented with a nonclassical inflammatory profile, whereas the mixed neutrophilic/mastocytic group revealed profiles consistent with both neutrophilic inflammation and hypersensitivity. Our adopted multi-omics approach provided a holistic assessment of the immunological status of the lower airways associated with the different cytological profiles of equine asthma.

## 1. Introduction

Airway inflammation is highly prevalent in racehorses, with the majority of non-infectious cases being defined as mild-to-moderate equine asthma (MMEA) [1,2,3,4]. Although a number of factors have been associated with the development of MMEA, an incomplete understanding of the precise cause and course of events underpinning this syndrome limits current treatment options [4]. Further understanding of the mechanisms underpinning MMEA offers the potential to identify specific targets for novel therapeutic and preventative interventions. Contrary to human asthma, where lung function testing is considered the gold standard diagnostic approach, the examination of airway secretions, particularly bronchoalveolar lavage fluid (BALF), is considered the primary ancillary diagnostic method in the diagnosis of equine MMEA [4]. In this respect, there is an ongoing need to identify sensitive and specific biomarkers that can be applied in a routine laboratory setting [4]. The identification of appropriate biomarkers would facilitate efforts to distinguish infectious and non-infectious lower airway inflammation and has the potential to unveil distinct pathological processes underpinning MMEA.

Although airway inflammation is an integral criterion in the MMEA clinical phenotype, there is a degree of inconsistency with respect to lower airway cytology. The differential cytology of BALF may reveal a predominantly neutrophilic, eosinophilic, metachromatic, or mixed inflammatory signature (MMEA phenotypes), a phenomenon likely to reflect a degree of aetio-pathogenic variability between cases. Indeed, defining MMEA beyond the current cytological phenotypes was a recently prioritized research aim, largely with a view to further elucidate the likely varied pathogeneses of this syndrome and identifying candidates that may benefit from a “precision medicine” therapeutic approach akin to specific human asthma endotypes [5,6,7]. The application of a complimentary multi-omics approach to comparative analyses between MMEA cytologic phenotypes has the potential to elucidate bespoke “type”-associated pathways, a prerequisite for the development of many novel and targeted therapeutic approaches. Prior attempts to define different MMEA cytological phenotypes based solely on the differential gene expression of selected cytokines yielded highly inconsistent and variable results [8,9,10,11], likely reflecting the complexity of, and overlap between, the aetio-pathogenetic pathways underpinning each phenotype. The inclusion of the evaluation of both global gene expression (RNA-seq) and peptide abundance in such a methodological approach significantly increases the potential to derive more extensive data from these sample sets and maximize the likelihood of identifying “endotype”-specific inflammatory pathways.

The aim of this study was to further define MMEA beyond the current cytological phenotypes, with a view to further elucidate the likely varied pathogenesis of this syndrome. We hypothesized that, for each cytological MMEA type, a multi-omics approach would reveal the key inflammatory pathways underpinning inflammatory cell recruitment to the lower airways and potentially reveal molecular targets that could be exploited in the design of novel prophylactic and therapeutic strategies.

## 2. Materials and Methods

### 2.1. Horses Used in This Study

A total of 27 French Trotters (17 males and 10 females; age mean: 4.1 + 0.4 (SEM) years; range: 1.8–7.7 years) were included in this study. Samples were obtained as part of a routine assessment of respiratory health (differential cytology), and residual samples were retained for the transcriptomic and proteomic analyses of this study. The Regional Ethic Committee for Clinical and Epidemiological Veterinary Research (CERVO-2020-3-V) approved all protocols involving animal use. Standard welfare procedures were followed, and informed owner consent was obtained for inclusion in the study.

Bronchoalveolar lavage fluid (BALF) samples from 19 horses were confirmed with airway inflammation (MMEA), as defined by differential BALF cytology. From those, four horses had a BALF cytology profile consistent with neutrophilic MMEA (Group_B_NEUT, *n* = 4), eight with mastocytic MMEA (Group_C_MAST, *n* = 8), four with combined neutrophilic and mastocytic MMEA (Group_D_NEUT_MAST, *n* = 4), and three with combined eosinophilic and mastocytic MMEA (Group_E_EOS_MAST, *n* = 3). Eight control samples were collected from horses in training to minimize any potential confounding effect of training per se [12]. Mild-to-moderate equine asthma was defined according to the American College of Veterinary Internal Medicine (ACVIM) guidelines by using the following BALF cell ratios in horses without systemic signs of disease or increased respiratory effort at rest: neutrophils > 5% and/or mast cells > 2% and/or eosinophils >1% [5,9,13,14].

### 2.2. Sample Collection

Prior to sample collection, the absence of clinical abnormalities was confirmed by the European College of Equine Internal Medicine diplomate specialists and highly experienced veterinarians in equine respiratory medicine. A total of 27 BALF samples from French Trotters were collected in France, as previously described [15]. Enrolled horses underwent physical examination and respiratory tract endoscopy. All methods reported in this study are in accordance with ARRIVE guidelines (https://arriveguidelines.org) [16].

A total of 300 microliters of fluid from BALF was cytocentrifuged (80 g, 10 min, Shandon Cytospin, Thermo Scientific, Wilmington, DE, USA) and stained with May-Grünwald-Giemsa. A differential cell count was performed on 300 cells, and the number of each cell type was recorded as a percentage of total nucleated cells, excluding epithelial cells [17,18]. Horses were considered free from MMEA based on the differential cell ratios not exceeding the following cut-off values: neutrophils: 5%; mast cells: 2%; eosinophils: 1% [5,9,13]. Based on these thresholds, the animals were categorized into five different groups, as shown in Table 1.

### 2.3. RNA Analysis of Equine Bronchoalveolar Lavage-Derived Cells

#### 2.3.1. Total RNA Extraction

A total of 25 milliliters of BALF were centrifuged at 400× *g* for 10 min, and the cell pellets were resuspended in 1 mL RNAprotect Cell Reagent (Qiagen, Courtaboeuf, France). Total RNA was extracted using RNAeasy plus micro kit (Qiagen, cat no 74034), according to the manufacturer’s instructions. gDNA Eliminator Spin Columns were used for genomic DNA removal. Following transfer to a clean tube for the precipitation step, 0.5 mL of 70% ethanol was added and then transferred to an RNeasy spin column and centrifuged at 18,000× *g* for 1 min at room temperature. Following centrifugation, the flow through was removed, and the RNA was washed once with RW1 buffer. The RNA membrane was then washed with RW1 and RPE. Finally, RNA was eluted in 30 µL RNase-free water, and RNA samples were stored at −80 °C until further use.

#### 2.3.2. RNA Quality Assessment

RNA concentration and purity were measured using an ND-1000 Nanodrop spectrophotometer (Thermo Scientific, Wilmington, DE, USA) by measuring absorbance at 260 and 280 nm (A260 and A280, respectively). The purity of RNA was determined using the A260/A280 ratio. A ratio close to 2 was considered to be indicative of pure RNA. RNA integrity was confirmed with the High Sensitivity RNA ScreenTape system (Agilent Technologies, Palo Alto, CA, USA). An RNA integrity number (RIN) greater than 7 was considered appropriate for RT-qPCR and RNA-seq analysis.

#### 2.3.3. RNA Library Preparation and NovaSeq Sequencing

RNA sequencing analysis was performed by Genewiz (Azenta Life Sciences, Frankfurt, Germany). Total RNA was processed to generate cDNA libraries and was subsequently sequenced using an Illumina NovaSeq platform (Illumina, San Diego, CA, USA) at a depth of 35 M reads strand-specific 150 bp paired-end per sample. Ribosomal RNA (rRNA) was depleted from samples for total RNA-seq. The RNA samples were quantified using Qubit 4.0 Fluorometer (Life Technologies, Carlsbad, CA, USA), and RNA integrity was checked using an RNA Kit on Agilent 5300 Fragment Analyzer (Agilent Technologies, Palo Alto, CA, USA). RNA sequencing library preparation was achieved using NEBNext Ultra II Directional RNA Library Prep Kit for Illumina following the manufacturer’s instructions (NEB, Ipswich, MA, USA). Briefly, the mRNAs were first enriched with Oligo (dT) beads. The enriched mRNAs were fragmented. The first strand and second strand of the cDNA were subsequently synthesized. The second strand of cDNA was marked by incorporating dUTP during the synthesis. The cDNA fragments were adenylated at 3′ends, and the indexed adapter was ligated to cDNA fragments. Limited cycle PCR was used for library amplification. The dUTP incorporated into the cDNA of the second strand enabled its specific degradation to maintain strand specificity. Sequencing libraries were validated using an NGS Kit on an Agilent 5300 Fragment Analyzer (Agilent Technologies, Palo Alto, CA, USA) and quantified by using Qubit 4.0 Fluorometer (Invitrogen, Carlsbad, CA, USA).

The sequencing libraries were multiplexed and loaded on the flow cell using an Illumina NovaSeq 6000 instrument according to the manufacturer’s instructions. The samples were sequenced using a 2 × 150 Pair-End (PE) configuration v1.5. Image analysis and base calling were conducted using NovaSeq Control Software v1.7 on a NovaSeq instrument. The raw sequence data (.bcl files) generated by Illumina NovaSeq were converted into fastq files and de-multiplexed using the Illumina bcl2fastq program version 2.20. One mismatch was allowed for index sequence identification.

#### 2.3.4. Processing of RNA Sequencing Data and Differential Expression Analysis

The raw data were deposited in Gene Expression Omnibus under the study accession number GSE277308. The sequence reads were trimmed to remove possible adapter sequences and poor-quality nucleotides using Trimmomatic v.0.36. The quality control of trimmed data was assessed using FASTQC [19]. The trimmed reads were mapped to the Equus caballus reference genome EquCab3.0 (available on ENSEMBL) using STAR aligner v.2.5.2b. STAR aligner is a splice-aware aligner that detects splice junctions and incorporates them to help align the entire read sequences. BAM files were generated as a result of this step. Unique gene hit counts were calculated by using Feature Counts from the Subread package v.1.5.2. Only uniquely mapped reads that fall within exon regions were counted.

After the extraction of gene hit counts, the gene hit counts table was used for downstream differential expression analysis. By using DESeq2, a comparison of gene expression between the groups of samples was performed. The Wald test was used to generate *p*-values and log2 fold changes. Genes with adjusted *p*-values or false discovery rate (FDR) ≤ 0.05 were identified as differentially expressed genes for each comparison. A PCA analysis was performed using the “plotPCA” function within the DESeq2 R package [20]. The plot shows the samples in a 2D plane spanned by their first two principal components. Analysis and data visualization were performed in R v 3.5.0.

#### 2.3.5. Quantitative Polymerase Chain Reaction (qPCR)

A total of 0.5 micrograms of total RNA was converted to complementary DNA (cDNA) using the prescription NanoScript reverse transcription kit (SuperScript III First-Strand Synthesis System, Invitrogen, Cat No 18080051, Waltham, MA, USA), according to the manufacturer’s instructions. The cDNA was stored at −20 °C until use. The transcript levels were calculated in triplicate using an MX3005P qPCR system (Stratagene) with the primers listed in Appendix A and qPCRBIO SyGreen Mix Lo-ROX kit (PCRBIO, London, UK). Primer efficiency was validated using a standard curve of five serial dilution points and *SDHA* as a housekeeping gene. *SDHA* was selected as a reference gene, as it remained stable in the RNA-seq data, and it has previously been evaluated as the most stable housekeeping gene to study equine exercise-induced stress data [5,21]. Reverse transcriptase and “no template” control samples were included in each run as negative controls. The data were analyzed using Stratagene MxPro v.4.10 software, and relative gene expression was calculated using the 2^−ΔΔCT^ method [22].

#### 2.3.6. Protein Analysis of Equine Bronchoalveolar Lavage-Derived Cells

Briefly, an aliquot of 500 µL of untreated BALF samples was homogenized in protein extraction buffer (100 mM Tris, pH 7.6, and 4% *w*/*v* SDS) + 1% Halt Protease Inhibitor Cocktail, EDTA-Free (Thermo Scientific™, Loughborough, UK, cat no 87785), as previously described [23]. Following homogenization, the samples were centrifuged at 20,000× *g* for 20 min at 10 °C. The supernatant containing the solubilized protein was removed and stored at −80 °C. The protein concentration of samples was determined using a Micro BCA Protein Assay Kit (Thermo Scientific™, Loughborough, UK, cat no 23235) according to the manufacturer’s instructions. Finally, total protein analysis was carried out for quality control purposes and to determine the equivocal protein load between samples. The samples were separated by electrophoresis on gradient gels (NuPAGE 4–12% Bis-Tris Protein Gels, 1.0 mm, 12-well, Fisher Scientific, cat no: NP0322BOX, Waltham, MA, USA) and stained with InstantBlue™ Protein Stain (Expedeon Ltd., cat no ISB1L, Cambridge, UK), as previously described [23,24]. The stained gel was then imaged using the LICOR Odyssey imager (LI-COR, Lincoln, NE, USA) to visualize and quantify the total protein load within each lane of the gel using the associated Image Studio Software (Version 5.2).

#### 2.3.7. S-Trap Proteolytic Digestion and LC-MS

A volume of 20 µg of each sample was used for the tryptic digestions. The samples were reduced with dithiothreitol and alkylated with iodoacetamide prior to tryptic digestion on S-TRAP (Protifi, Fairport, NY, USA) cartridges, following standard protocol [25]. The resulting peptides were cleaned up using C_18_ stage tips. Purified peptides were separated over a 90 min gradient on an Aurora-25 cm column (IonOpticks, Victoria, Australia) using an UltiMate RSLCnano LC System (Dionex) coupled to a timsTOF FleX mass spectrometer (Bruker Daltonics, Bremen, Germany) through a CaptiveSpray ionization source. The gradient was delivered at a flow rate of 200 nL/min, and washout and equilibration were performed at 500 nL/min. The column temperature was set at 50 °C. For DDA-PASEF acquisition, the full scans were recorded from 100 to 1700 m/z, spanning from 1.45 to 0.65 Vs/cm^2^ in the mobility (1/K0) dimension. Up to 10 PASEF MS/MS frames were performed on ion-mobility-separated precursors, excluding singly charged ions, which are fully segregated in the mobility dimension, with a threshold and target intensity of 1750 and 14,500 counts, respectively. The raw mass spectral data were processed using PEAKS Studio version X-Pro Software (Bioinformatics Solutions Inc., Columbia, ON, Canada). A search was conducted against the equine (*Equus caballus*) sequence database (UniProt Proteome ID: UP000002281), which contains 20,865 entries. The MS1 precursor mass tolerance was set to 20 ppm, and the MS2 fragment ion tolerance was 0.06 Da. The search parameters specified fully tryptic digestion, allowing for one missed cleavage. Cysteine was treated as a fixed modification with a mass addition of [+57.02], while methionine oxidation and the deamination of asparagine and glutamine were set as variable modifications. Quantitative LFQ analysis was performed using default parameters with optional ID transfer enabled.

#### 2.3.8. Statistical Analysis of Proteomic Data

The proteomic data were processed, as previously described [26]. Briefly, the data were transformed into a logarithmic scale, normalized, and imputed for missing values, and fold change was generated. The Shapiro–Wilk test was performed to assess whether the replicate values were normally distributed. If significant for at least one group (*p*-value ≤ 0.05), the given protein in that condition did not have normally distributed data. When both conditions were normally distributed, a *t*-test was performed. When data were not normally distributed based on the Shapiro–Wilk test, the nonparametric Mann–Whitney test was applied. Statistical significance was assumed at *p* < 0.05. To select the type of *t*-test performed (homoscedastic or heteroscedastic), an F-test was used to check whether the data were homoscedastic or heteroscedastic, and then a two-sample equal variances (homoscedastic) or two-sample unequal variances (heteroscedastic) *t*-test was performed, respectively. If the data were homoscedastic, there was a small variance between the replicates in both groups (F ≥ 0.05). If the data were heteroscedastic, there was a big variance between replicates in both groups (F < 0.05).

#### 2.3.9. Gene Ontology and Pathway Analysis

The identification of enriched biological processes and KEGG pathways in the upregulated and downregulated gene/protein lists was performed using the Database for Annotation, Visualization, and Integrated Discovery (DAVID) database for Gene Ontology (GO) with Knowledgebase v2024q2) [27,28,29]. To gain a better view of the results, functional analysis was also performed, as previously described, using Ingenuity Pathway Analysis to infer the functional roles and relationships of the differentially expressed genes based on the log2 fold change value of each molecule [12,30].

## 3. Results

The aims of this project were three-fold: (1) to establish the conditions for optimal RNA and protein isolation from equine BALF samples, (2) to perform RNA-seq and proteomic analysis on the BALF samples, providing a data resource for the community, and (3) to perform downstream analyses on the respective datasets to reveal MMEA endotype-specific pathways. The bronchoalveolar lavage samples provided good quality and yield regarding both RNA and protein for downstream transcriptomic/proteomic analyses, as previously described [23,31].

### 3.1. Differential Cell Count of Bronchoalveolar Lavage Samples

The differential cell counts of the BALF samples are presented in Figure 1. In line with common practice in both human and equine pulmonology, the epithelial cells were excluded from the differential cell count [32,33], an approach largely justified by the extensive variability in their proportion relative to other cell types and the potential for their inclusion to significantly skew data derived from downstream analyses [33]. The variability in epithelial cell proportion can be influenced by factors such as coughing and sample collection techniques. In agreement with previous studies, a remarkable percentage of BALF cells were macrophages (including hemosiderophages) [23,31,34]. As all samples were derived from racehorses, the presence of hemosiderophages was expected due to the high prevalence of exercise-induced pulmonary hemorrhage in this population [34].

### 3.2. Total RNA and Protein Extraction of Bronchoalveolar Lavage Samples Was Successfully Performed

Bronchoalveolar lavage samples from 27 French Trotters were collected, total BALF cells were isolated, and an RNA extraction protocol was performed, as previously described [18]. The average RNA yield extracted from the 8 × 10^6^ BALF cells was 75.8 + 8.7 ng/µL. The RNA samples derived from the BALF cells had an average RIN number of 9 + 0.1 (SEM), exceeding the threshold of 7, which is recommended for RNA-seq and qPCR analysis. RNA was submitted for RNA-seq analysis at Genewiz from Azenta Life Sciences. Genewiz also performed quality control prior to DNA strand-specific library preparation. The subsequent sequencing was conducted based on an Illumina platform using 150 bp paired-end sequencing (35 M coverage; 35 million reads generated through the sequencing process). The selected genes were evaluated using qPCR to confirm their differential expression (Appendix A).

In order to expand on the RNA studies, we also defined the airway protein profiles (total proteome), thus revealing the mechanisms that may underpin any alterations in immune function between different cytological profiles. Protein extraction was successfully performed on the BALF samples from 27 horses. An average of 0.21 ± 0.33 (±SEM) mg of protein was isolated from 500 µL of BALF per animal. The results are consistent with previous studies [23]. To visualize the total protein load, all samples were run on gradient gels and stained with instant blue protein stain, as previously described [23]. Figure 2 is representative of a gel stain of the 11 equine BALF samples. The bands show a similar pattern across the samples, as previously reported [23]. The protein samples were subsequently submitted for proteomic analysis.

### 3.3. Distinct Molecular Pathways Underpinning Mild-To-Moderate Equine Asthma Cytological Profiles

#### 3.3.1. Molecular Profile of Racehorses with High BALF Neutrophil Ratios

Whole-transcriptome (RNA-seq) and proteome profiling was performed on the BALF cells and BALF (respectively) derived from racehorses with neutrophil ratios exceeding 5% (Group_B_NEUT; *n* = 4) versus healthy individuals (Group_A_Control; *n* = 8).

*RNA-seq:* In total, 13,059 equine genes were identified, and the reads were quantified to identify those that were differentially expressed between the two groups. A list of 17 differentially expressed transcripts (following a false discovery rate (FDR) of ≤0.05) was detected between Group_B_NEUT and Group_A_Control (Figure 3: the full list is in Appendix A). From those, 12 genes were upregulated, and five were downregulated. The differentially expressed genes were related to the regulation of epithelial cell proliferation or response to stimulus; these included *EQMHCC1*, *GSTA4*, *CCND1*, and *THBS1*.

*Proteome:* In total, 1724 unique proteins were detected, 215 of which were identified as differentially expressed based on a *p*-value of ≤0.05. By applying these criteria, 157 proteins were upregulated, and 58 were downregulated. The two groups showed distinct patterns of gene/protein expression in both datasets (Figure 3A,C; the full list is in Appendix A).

*Pathway analyses:* An analysis of the KEGG pathway and biological process enrichment of the upregulated and downregulated proteins was performed using DAVID software v2024q2 [27]. The analysis revealed the pathways and biological processes involved in immune defense and immune system process, as well as oxidative stress and metabolic processes (Appendix A). To gain a better view of the results, functional analysis was also performed, as previously described, using IPA [12]. As expected, the analysis of samples derived from animals with high neutrophil counts was consistent with biological processes related to neutrophil chemotaxis, phagocytosis, and inflammation of the respiratory system (Figure 4).

#### 3.3.2. Molecular Profile of Racehorses with High Mast Cell Counts in the BALF Samples

Whole-transcriptome (RNA-seq) and proteome profiling was performed on the BALF cells and BALF (respectively) derived from racehorses with BALF mast cell ratios exceeding 2% (Group_C_MAST; *n* = 8) versus healthy individuals (Group_A_Control; *n* = 8).

RNA-seq: A total of 13,331 equine genes were identified, and the reads were quantified to identify those differentially expressed between the two groups (Figure 5B; the full list is in Appendix A). Mast cell influx in the lower airways was associated with a change in the equine gene expression of 21 differentially expressed genes (19 upregulated and two downregulated). We defined differentially expressed genes as those showing up or downregulation with a false discovery rate (FDR) below 0.05. As expected, genes that play an important role in mast cell degranulation and mast cell-mediated immunity (FCER1A and KIT) were significantly upregulated in horses with high mast cell ratios compared to controls. Others were related to cell communication and cell signaling (*MS4A2*, *PDE1C*, *PTPRM*, *RGS13*, *RET*).

Proteome: From the 1724 unique proteins detected, 359 were identified as differentially expressed, based on a *p*-value of ≤0.05. From these, 213 were upregulated, and 146 were downregulated (Figure 5D; the full list is in Appendix A).

*Pathway analyses:* The enrichment analysis performed using DAVID annotation software v2024q2 revealed the biological processes related to oxidative stress, apoptosis, and cellular metabolic processes (Appendix A). The pathway analysis was also complemented using IPA software v24.0.1. In contrast to the comparisons made with Group B NEUT, which revealed the processes related more to acute phase response, inflammation, and neutrophil activation, the samples from Group_C_MAST were enriched with biological processes related to hypersensitivity reaction, inhibition of airway inflammation, and fibrosis (Figure 6).

#### 3.3.3. Molecular Profile of Racehorses with Combined High BALF Neutrophil and Mast Cell Ratios

Whole-transcriptome (RNA-seq) and proteome profiling was performed on the BALF cells and BALF (respectively) derived from racehorses with combined BALF neutrophil ratios exceeding 5% and mast cell ratios exceeding 2% (Group_D_NEUT_MAST; *n* = 4) versus healthy individuals/controls (Group_A_Control; *n* = 8).

*RNA-seq*: A total of 13,002 equine genes were identified, and the reads were quantified to identify those differentially expressed between the two groups (Figure 7B; the full list is in Appendix A). Neutrophil and mast cell influx in the lower airways was associated with a change in the expression of 31 differentially expressed genes (22 upregulated and nine downregulated). Differentially expressed genes were defined based on an FDR of ≤0.05. Consistent with the sample cytology, genes related to neutrophil extracellular trap formation (H2BC4, H4C3, and H4C4) and neutrophil chemotaxis (*TREM1*) were significantly upregulated in this group [35].

*Proteome*: In total, 1724 unique proteins were detected, 299 of which were identified as differentially expressed, based on a *p*-value of ≤0.05. Of these, 146 were upregulated, and 153 were downregulated (Figure 7D; the full list is in Appendix A).

*Pathway analyses:* The identification of enriched KEGG pathways and biological processes in the upregulated and downregulated protein lists performed using DAVID annotation software v2024q2 is shown in Appendix A. The detected protein list included molecules involved in neutrophil chemotaxis, airway inflammation, and hypersensitivity (Figure 8), as indicated by the IPA analysis. This finding was highly consistent with the increased airway neutrophil and mast cell ratios that defined this group.

#### 3.3.4. Molecular Profile of Racehorses with High Eosinophil and Mast Cell Ratios on BALF Samples

Whole-transcriptome (RNA-seq) and proteome profiling was performed on the BALF cells and BALF (respectively) derived from racehorses with high BALF eosinophil ratios exceeding 1% and mast cell ratios exceeding 2% (Group_E_EOS_MAST; *n* = 3) versus healthy individuals/controls (Group_A_Control; *n* = 8).

*RNA-seq:* A total of 13,360 equine genes were identified, and the reads were quantified to identify those that were differentially expressed between the two groups (Figure 9B; the full list is in Appendix A). Eosinophil and mast cell influx in the lower airways was associated with a change in expression of 34 differentially expressed genes (24 upregulated and 10 downregulated). Differentially expressed genes were defined based on an FDR of ≤0.05.

*Proteome*: In total, 1724 unique proteins were detected, 225 of which were identified as differentially expressed, based on a *p*-value of ≤0.05, of which 136 proteins were upregulated, and 89 were downregulated (Figure 9C,D; the full list is in Appendix A).

*Pathway analyses:* To gain a global view of the results, functional analysis was performed, as previously described, using IPA and DAVID (Appendix A). In this group, several proteins involved in an alternative polarization profile (M2) were detected using IPA (Figure 10).

## 4. Discussion

Airway inflammation is an integral component of the MMEA clinical phenotype; however, there is a degree of inconsistency with respect to the lower airway cytology profile applied to fulfill this diagnostic criterion. From an inflammatory cell perspective, the differential cytology of BALF from MMEA cases may be exclusively neutrophilic, eosinophilic, or metachromatic, or a mixed population of inflammatory cells may be detected. This phenomenon is highly consistent with the degree of aetio-pathogenic variability between MMEA subtypes. Such variability warrants further investigation with a view to identifying novel and more targeted therapeutic and/or preventative strategies. In line with such potential benefits, defining MMEA beyond the currently restricted cytological phenotypes was recently identified as a prioritized research aim [4,14,36,37]. To the best of our knowledge, this is the first report describing the application of a complimentary multi-omic methodological approach to dissect the pathogenesis of different MMEA endotypes in racehorses. Indeed, specific gene and protein signatures were defined between the different groups, and specific pathways in relation to each cytological profile were discovered. Furthermore, this analytical process has revealed some similarities with the variable pathogenesis of different human asthma endotypes [38].

Overall, the number of the differentially expressed genes detected in the intergroup comparisons was similar to those previously reported by others and ourselves, using similar transcriptomic technologies [5,12,39]. Although the validity of the data, in terms of differential expression, was supported by the qPCR results, the relatively low inflammatory cell ratio cut-offs applied as group inclusion criteria (mast cells > 2%; eosinophils > 1%; neutrophils > 5%) likely limited the depth of information that could be derived from the analyses. This limitation was likely most applicable to the transcriptomic data, which were derived from the entire BAL fluid cell population, thus masking small (but potentially biologically relevant) intergroup differences in gene expression. Small group sizes may also have been a limiting factor in the group-associated differential expression analyses due to the potential bias of a single individual on the collective group gene transcription data. This may partly explain the more limited level of differential expression derived from the gene lists compared to the more extensive protein lists, a factor that could not be attributed to RNA integrity, as the RIN numbers in the extracted RNA exceeded a threshold of 7. Furthermore, the statistical analyses applied to RNA-seq data are considered more stringent (*p*-adjusted/FDR) than those applied to proteomic data (*p*-value), where corrections for multiple testing are not recommended [20,26]. Despite the above limitations and the relative clinical quiescence of MMEA compared with other airway diseases (e.g., severe equine asthma), specific MMEA-type pathway patterns were overt. This is promising from the perspective of both diagnostic biomarkers and therapeutic target discovery.

Horses with neutrophilic inflammation showed enrichment in different inflammatory processes with the differential expression of several inflammatory proteins (ELANE, CD14, APOE, LYZ) and neutrophil attractants (CD14, ELANE, ICAM1, MPO, LYZ) reaching statistical significance (Appendix A). Although other inflammatory transcripts (*CCL11*, *CXCL6*, *MUC5AC*) and neutrophil attractants (*CSF3R*, *CXCL1*, *CXCL6*, *VCAM1*, *IL13*) were also highly expressed in this group, they failed to meet the preset statistical significance threshold, likely resulting from some of the limitations described above. From a mucus production perspective, the upregulation of *EGFR* (even though it did not pass the significance threshold) has been shown to be involved in mucin production by airway goblet cells in asthma [40]. The upregulation of the mucin genes *MUC5AC*, *MUC20*, and *MUC1* in the neutrophilic group (Appendix A) also aligns with previous reports, highlighting the differential expression of a number of inflammatory genes in horses with neutrophilic BALF cytology [5]. Moreover, IL6-activated neutrophils have been shown to augment the neutrophilic release of elastase (ELANE; Elastase, Neutrophil Expressed) [41]. Notably, ELANE was significantly upregulated in the proteomic list of this MMEA subgroup. The IL6 stimulation of neutrophils is known to regulate the neutrophil production of mediators such as platelet-activating factor (PAF) and reactive oxygen species (ROS) [42], thus aligning with the high enrichment of platelet-activating factor acetylhydrolase 1b catalytic subunit 3 (PAFAH1B3), IL6ST, and ROS in the proteome of this MMEA subgroup. Interestingly, IL6 is considered a potential biomarker for non-Th2 asthma in humans [43].

In comparison to the neutrophilic cases, the IPA analysis of the proteomic datasets derived from horses with metachromatic inflammation revealed enrichment in pathways related to *hypersensitivity reaction* (CD44, ICAM1, SOD1, PSAP, CDH1) and *lung fibrosis* (AGER, TGFBR2, FBLN1, S100A9) (Figure 6). These findings are highly consistent with previous observations, suggesting the important role of mast cell-derived mediators in the pathogenesis of airway remodeling in horses [5]. In particular, mast cells tend to accumulate in lung compartments, such as the alveolar parenchyma, bronchial epithelium, and smooth muscle, of patients with allergic and severe asthma, thereby presumably increasing the detrimental consequences of mast cell activation in these cases, which entail airway remodeling and hypersensitivity reactions [44,45].

Unsurprisingly, the molecules involved in mast cell degranulation and mast cell-mediated immunity (*FCER1A*, *KIT*, CCL8, TGFBR2) were significantly upregulated in horses with the metachromatic MMEA subtype (Appendix A). These findings align with those of Davis and Sheats (2021), who reported an upregulation of *KIT* mRNA expression in BALF cells from horses with astrocytic airway inflammation [5]. Interestingly, *KIT* is a member of the receptor tyrosine kinase III family and a master regulator of the mast cell lineage [46]. *KIT*, validated here using qPCR results, is also found to play an important role in lung hypersensitivity, fibrosis, and the Th2-high human asthma endotype [46,47]. In line with these observations and in contrast to the data derived from the neutrophilic group, the “*inflammation of the respiratory system component*” IPA-generated network was found to be inhibited in the metachromatic group. This finding suggests that mast cells may induce a nonclassical inflammatory effect in the lung (Figure 6) and highlights their important role in asthma [48,49]. Indeed, mast cell accumulation in the lung is extensively considered a pathological feature of human allergic asthma, in which Th2 differentiation is preferentially promoted or supported [44,50,51]. Indeed, KIT, implicated in the human “Th2 high” asthma endotype, has recently been advocated as an appropriate therapeutic target molecule in cases of severe refractory asthma (e.g., imatinib or anti-KIT monoclonal antibodies) [46,47,52]. In light of the proteomic data derived from this study, the feasibility of a similar approach in refractory MMEA cases may be worth some consideration should further work support this hypothesis. Finally, the enrichment of proteins related to response to oxidative stress (PARK7, APOE, GCLC, GPX2, PRDX3, ROS1), cell redox homeostasis (GCLC, PRDX3, PRDX6), and oxidative phosphorylation (ATP5F1D, ATP5MG, ATP6V1G1, NDUFS3, COX6B1) (Appendix A) in the metachromatic group aligns with some of the mechanisms known to underpin mast cell degranulation in asthma-associated pathology in humans; namely, the overproduction of ROS with subsequent redox imbalance and oxidative stress [48,53].

Interestingly, the IPA-generated network analysis of the proteome also revealed the inhibition of “*inflammation of the respiratory system component*” in horses with a combined mast cell and eosinophilic BALF cytology profile (Figure 10), further supporting the role of mast cells in this inhibitory effect. This hypothesis is further complemented by the M2 polarization profile (APOE, TXN, TGM2, STING1, NAMPT) enriched in this group, a dominant macrophage profile in high Th2-type asthma in humans and one known to activate the Th2 response with downstream eosinophilic infiltration and airway remodeling [54].

The analysis of the samples derived from horses with a mixed immune cell infiltration revealed a combination of enrichment pathways related to the different immune cell roles (neutrophils, mast cells, and eosinophils). Such a mixed inflammatory response, characterized by a mixed population of immune cells within the airways, is also reported in human asthmatic patients [43]. Specifically in our study, samples characterized by a combination of high neutrophil and mast cell ratios showed an alteration in the expression of several molecules related to both airway inflammation (HSPA5, CYBB, ATP5F1A, PTPRC) and hypersensitivity (RAB27A, ALCAM, PTPRC, LYN). In contrast to the metachromatic group, KIT was not differentially expressed in the mixed neutrophilic/mast cell group; however, the involvement of KIT signaling in this cohort was supported by the increased expression of LYN, a tyrosine kinase protein implicated in mast cell activation [46]. Expectedly, proteins related to neutrophil recruitment (LPO, ITGB1, SYK) were also differentially expressed in this group compared to controls (Figure 8).

Overall, BALF cells from racehorses with neutrophilic inflammation showed enrichment in pathways involved in inflammatory reactions, while those from the metachromatic group showed enrichment in pathways involved in oxidative stress, hypersensitivity reaction, and tissue structure alterations, such as lung fibrosis. The level of agreement with a previous RNA-seq study on different MMEA subtypes is encouraging in light of the inconsistency and variability in the data derived from previous studies adopting an approach solely focused on the gene expression profiles of specifically selected cytokines [5,8,9,10,11,37]. This level of inconsistency derived from this restricted methodological approach and the level of agreement between studies adopting a global gene expression approach likely reflects the complexity of, and overlap between, the respective aetio-pathogenic pathways associated with each MMEA phenotype.

Our results support the adoption of a similar simple categorization approach applied to human asthma endotypes, namely, a “Type 2” or “non-Type 2” inflammatory type [55]. The neutrophilic MMEA cases in this study mimic a non-Th2 inflammatory profile, while the metachromatic or eosinophilic cases showed more similarity with a Th2 pattern (M2 polarization). Interestingly, human athletes with non-Type 2 asthma have significantly fewer allergic symptoms and fewer previously diagnosed allergic rhinitis episodes, suggesting the role of intensive athletic training per se in the development of this endotype [55]. Although we have previously reported on enriched biological processes relating to inflammation in airway-derived cells from horses in race training [12,31,56], this effect was unlikely to have influenced the inter-group comparisons made in the current study, as all samples were obtained from racehorses in training. Similarly, the immunomodulatory effect of exercise is well-recognized in humans. Furthermore, it has been suggested that asthma among endurance athletes could arise from hyperventilation and mechanical stress on the airway surfaces over prolonged periods, ultimately resulting in airway remodeling [57]. Consistent with this hypothesis and in line with the high prevalence of neutrophilic MMEA in performance horses, the prevalence of non-Type 2 asthma is remarkably high in human athletes, at a level that would not be expected in such a young population [55]. The influence of prolonged hyperventilation and associated shear stress has not been fully explored in the equine athlete and warrants further investigation.

## 5. Conclusions

This project used state-of-the-art investigative approaches to improve our understanding of the mechanisms underpinning a highly prevalent, performance-limiting disorder of racehorses: MMEA. This is a necessary step in the development of more targeted preventative and treatment options for this disorder. With the current study, we have expanded on previous methodological approaches by adopting techniques that permit consideration of global gene expression (RNA-seq) and proteomic analysis. This has significantly expanded the volume of information derived from these sample sets and maximized our ability to identify “endotype”-specific pathways with the potential to reveal appropriate diagnostic biomarkers and/or therapeutic targets. In its current holistic format, the study is of direct relevance to racehorses. The results derived from individual components of the study could be translationally applied to other breeds (Thoroughbreds), sporting disciplines (where MMEA is recognized), and even species (humans/elite athletes).

Current observations support the hypothesis that the different BALF cytological profiles reflect different MMEA transcriptomic and proteomic profiles and, thus, further inform efforts to establish specific MMEA disease endotypes. This study also highlights the value of the horse as an appropriate animal model with potential translational applications to human exercise immunology. There remains a requirement to expand this multi-omic analytical approach to a larger sample population to enable a more critical assessment of the gene/protein expression profile of MMEA endotypes. Finally, additional novel, cutting-edge technologies (e.g., single-cell sequencing and spatial transcriptomics) may further help to reveal disease- and/or disease susceptibility-associated biomarkers and/or novel therapeutic targets for both horses and humans.

## Figures and Tables

**Figure 1 cells-13-01926-f001:**
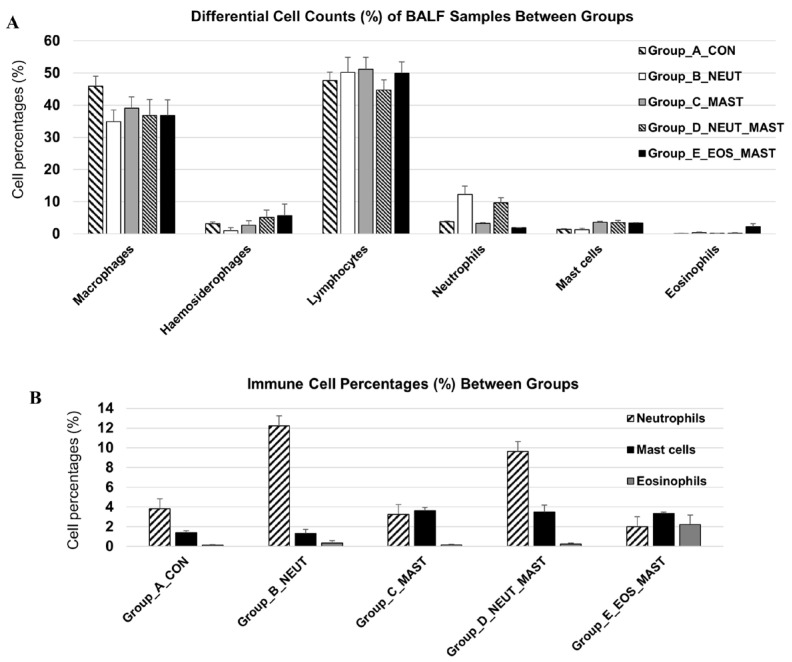
Bronchoalveolar lavage fluid cytological profile. Differential cell count (mean + SEM%) of equine BALF samples (*n* = 27) between the different groups (**A**) and the percentages of inflammatory cells of interest only (neutrophils, mast cells, and eosinophils) (**B**). Differential leucocyte count (minimum of 300 cells) was performed and expressed as a percentage of the total nonsquamous and non-epithelial nucleated cells.

**Figure 2 cells-13-01926-f002:**
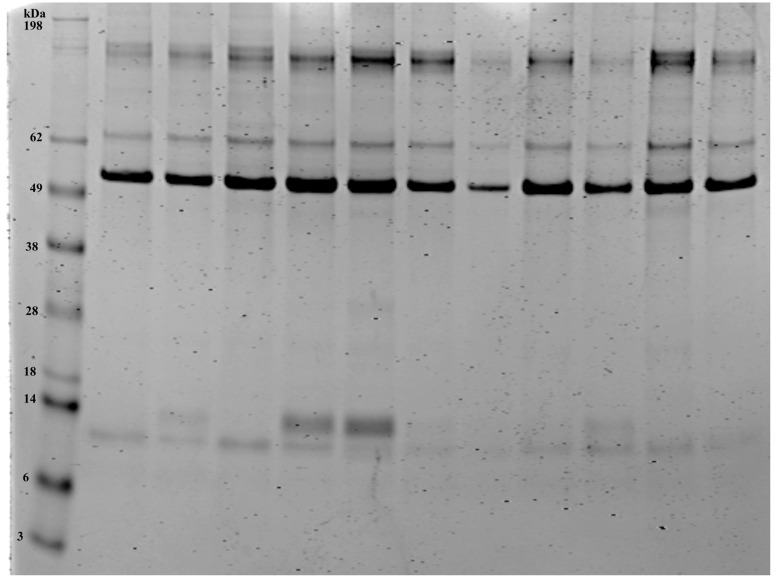
Total protein stain from the equine bronchoalveolar lavage samples. Lane 1 shows the protein ladder. Total protein stain of the equine BALF samples (Lanes 2–12, 5 µg). Note the diversity of proteins in the BALF samples. The stained gel was imaged using the LICOR Odyssey imager and the associated Image Studio Software, Version 5.2.

**Figure 3 cells-13-01926-f003:**
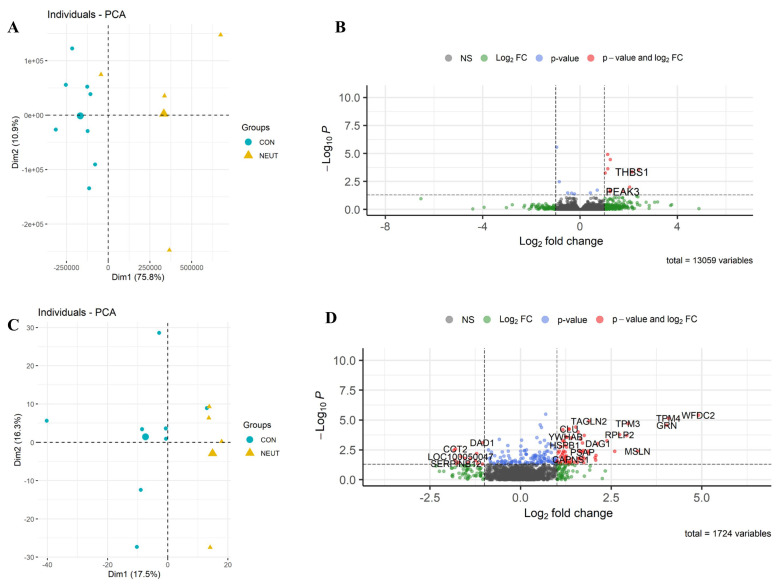
Differential expression of Group_B_NEUT. Scatterplot of the first two principal components (Dim1 and Dim2) of the RNA-seq (**A**) and proteomic samples (**C**) clustered according to their gene and protein expression; in parenthesis: original variance explained by each principal component. The larger symbols represent group means. Volcano plots of differentially expressed genes (**B**) and proteins (**D**), identified between Group_B_NEUT (NEUT) and the control group (CON). The green dots denote molecules with an absolute log2-fold change of >1. The blue dots denote genes with FDR ≤ 0.05, and the red dots denote those with FDR ≤ 0.05 and an absolute log2-fold change of >1 (**B**). In the proteomic dataset, the blue dots denote those with a *p*-value of ≤0.05, and the red dots denote those with a *p*-value of ≤0.05 and an absolute log2-fold change of >1 (**D**). Finally, the grey dots denote gene/protein expression without marked differences.

**Figure 4 cells-13-01926-f004:**
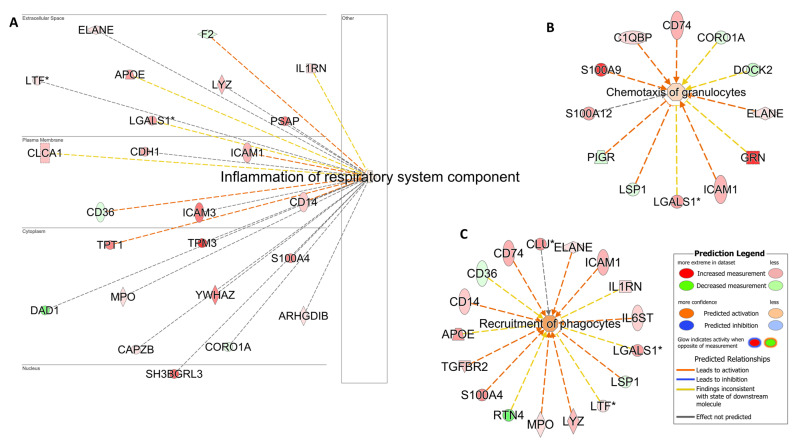
Airway inflammation and phagocyte cell activation. Pathway analysis identifies networks comprising *airway inflammation* (**A**), *Chemotaxis of granulocytes* (**B**) and (**C**) *Recruitment of phagocytes*. The biological processes and regulators are colored by their predicted activation state: activated (orange) or inhibited (blue). Darker colors indicate higher scores. The edges connecting the nodes are colored orange when leading to the activation of the downstream node, blue when leading to its inhibition, and yellow if the findings underlying the relationship are inconsistent with the state of the downstream node. The pointed arrowheads indicate that the downstream node is expected to be activated if the upstream node connected to it is activated, whereas the blunt arrowheads indicate that the downstream node is expected to be inhibited if the upstream node that connects to it is activated. The molecules in green are downregulated, and those in red are upregulated. The asterisk (*) indicates that multiple identifiers map to the molecule. The analysis was performed using the Ingenuity Pathway Analysis software v24.0.1.

**Figure 5 cells-13-01926-f005:**
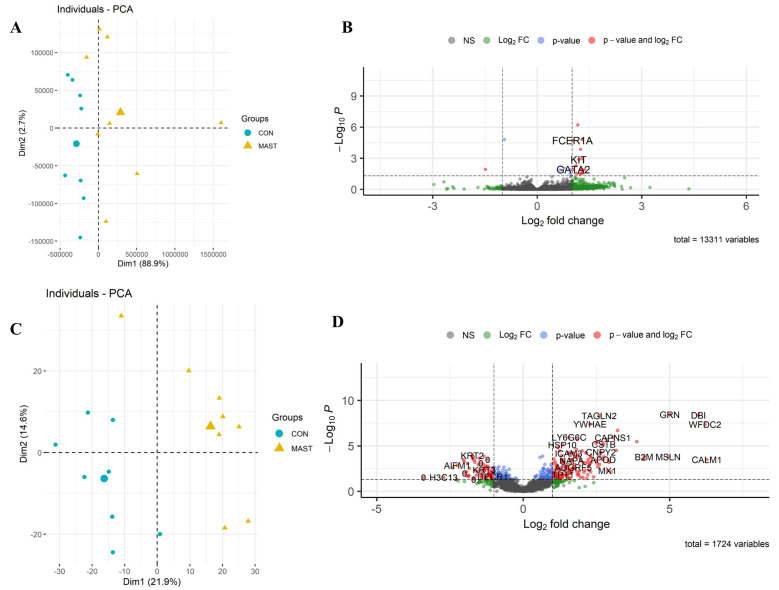
Differential expression of Group_C_MAST. Scatterplot of the first two principal components (Dim1 and Dim2) of the RNA-seq (**A**) and proteomic samples (**C**) clustered according to their gene and protein expression; in parenthesis: the original variance explained by each principal component. The larger symbols represent group means. Volcano plots of differentially expressed genes (**B**) and proteins (**D**), identified between Group_C_MAST (MAST) and the control group (CON). The green dots denote molecules with an absolute log2-fold change of >1. The blue dots denote genes with an FDR of ≤0.05, and the red dots denote those with an FDR of ≤0.05 and an absolute log2-fold change of >1 (**B**). In the proteomic dataset, the blue dots denote those with a *p*-value of ≤0.05, and the red dots those with a *p*-value of ≤0.05 and an absolute log2-fold change of >1 (**D**). Finally, the grey dots denote the gene/protein expression without marked differences.

**Figure 6 cells-13-01926-f006:**
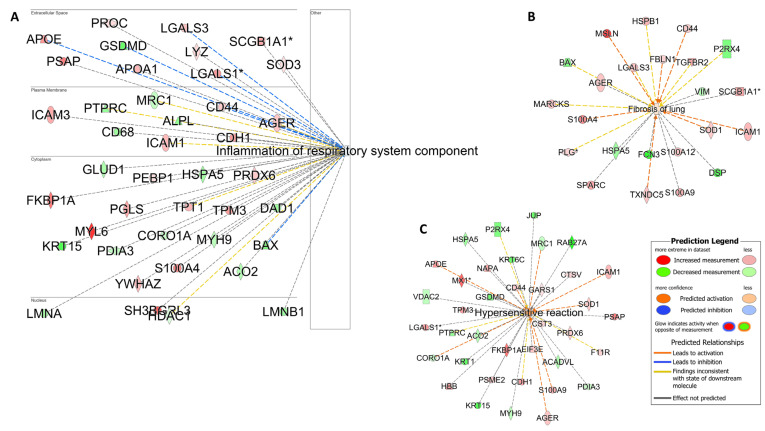
Hypersensitivity and fibrosis. Pathway analysis identifies networks comprising *airway inflammation* (**A**). *Fibrotic processes* (**B**) and *hypersensitivity reactions* (**C**) were also enriched in racehorses with high mast cell counts. The biological processes and regulators are colored according to their predicted activation state: activated (orange) or inhibited (blue). The darker colors indicate higher scores. The edges connecting the nodes are colored orange when leading to the activation of the downstream node, blue when leading to its inhibition, and yellow if the findings underlying the relationship are inconsistent with the state of the downstream node. The pointed arrowheads indicate that the downstream node is expected to be activated if the upstream node connected to it is activated, whereas the blunt arrowheads indicate that the downstream node is expected to be inhibited if the upstream node that connects to it is activated. The molecules in green are downregulated, and those in red are upregulated. The asterisk (*) indicates that multiple identifiers map to the molecule. The analysis was performed using the Ingenuity Pathway Analysis software v24.0.1.

**Figure 7 cells-13-01926-f007:**
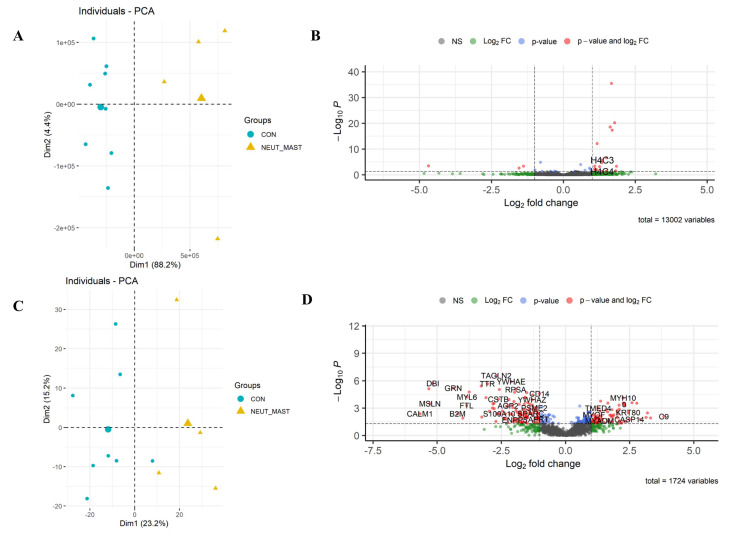
Differential expression of Group_D_NEUT_MAST. Scatterplot of the first two principal components (Dim1 and Dim2) of the RNA-seq (**A**) and proteomic samples (**C**), clustered according to their gene and protein expression; in parenthesis: the original variance explained by each principal component. The larger symbols represent group means. Volcano plots of the differentially expressed genes (**B**) and proteins (**D**) identified between Group_D_NEUT_MAST and the control group. The green dots denote molecules with an absolute log2-fold change of >1. The blue dots denote genes with an FDR of ≤0.05, and the red dots those with an FDR of ≤0.05 and an absolute log2-fold change of >1 (**B**). In the proteomic dataset, the blue dots denote those with a *p*-value of ≤0.05, and the red dots those with a *p*-value of ≤0.05 and an absolute log2-fold change of >1 (**D**). Finally, the grey dots denote the gene/protein expression without marked differences.

**Figure 8 cells-13-01926-f008:**
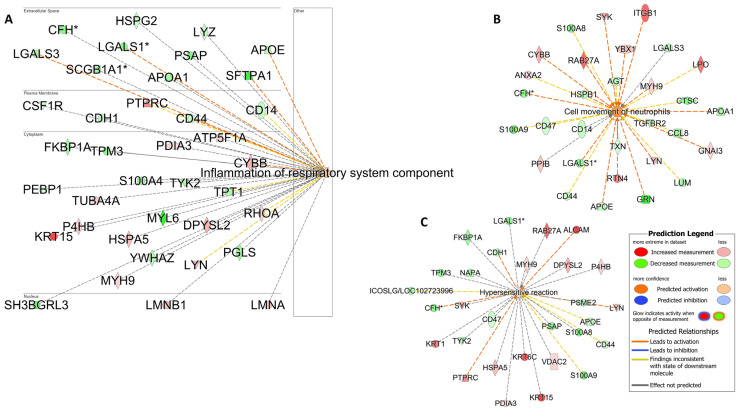
Airway inflammation and hypersensitivity reaction. Pathway analysis is used to identify networks comprising airway inflammation (**A**). Neutrophil migration (**B**) and hypersensitivity reactions (**C**) were also enriched in racehorses with high neutrophil and mast cell counts. The biological processes and regulators are colored by their predicted activation state: activated (orange) or inhibited (blue). The darker colors indicate higher scores. The edges connecting the nodes are colored orange when leading to the activation of the downstream node, blue when leading to its inhibition, and yellow if the findings underlying the relationship are inconsistent with the state of the downstream node. The pointed arrowheads indicate that the downstream node is expected to be activated if the upstream node connected to it is activated, whereas the blunt arrowheads indicate that the downstream node is expected to be inhibited if the upstream node that connects to it is activated. The molecules in green are downregulated, and those in red are upregulated. The asterisk (*) indicates that multiple identifiers map to the molecule. Analysis was performed using the Ingenuity Pathway Analysis software v24.0.1.

**Figure 9 cells-13-01926-f009:**
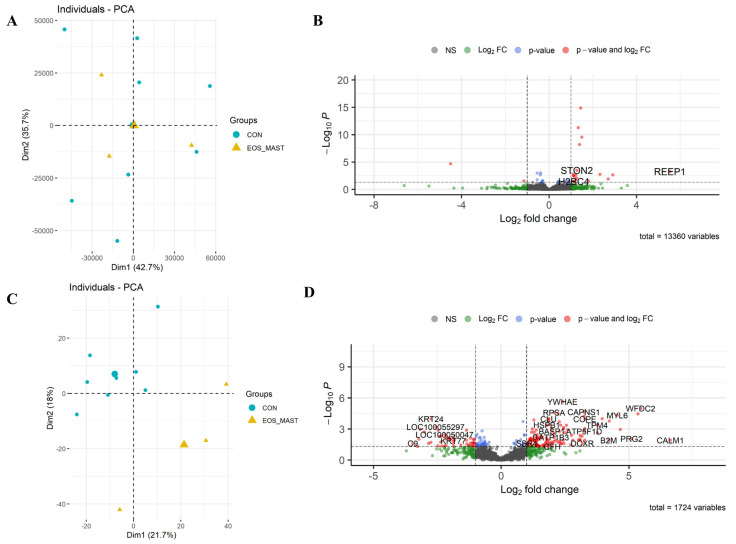
Differential expression of Group_E_EOS_MAST. Scatterplot of the first two principal components (Dim1 and Dim2) of the RNA-seq (**A**) and proteomic samples (**C**), clustered according to their gene and protein expression; in parenthesis: the original variance explained by each principal component. The larger symbols represent group means. Volcano plots of differentially expressed genes (**B**) and proteins (**D**), identified between Group_E_EOS_MAST and the control group. The green dots denote molecules with an absolute log2-fold change of >1. The blue dots denote genes with an FDR of ≤0.05, and the red dots those with an FDR of ≤0.05 and an absolute log2-fold change of >1 (**B**). In the proteomic dataset, the blue dots denote those with a *p*-value of ≤0.05, and the red dots those with a *p*-value of ≤0.05 and an absolute log2-fold change > 1 (**D**). Finally, the grey dots denote the gene/protein expression without marked differences.

**Figure 10 cells-13-01926-f010:**
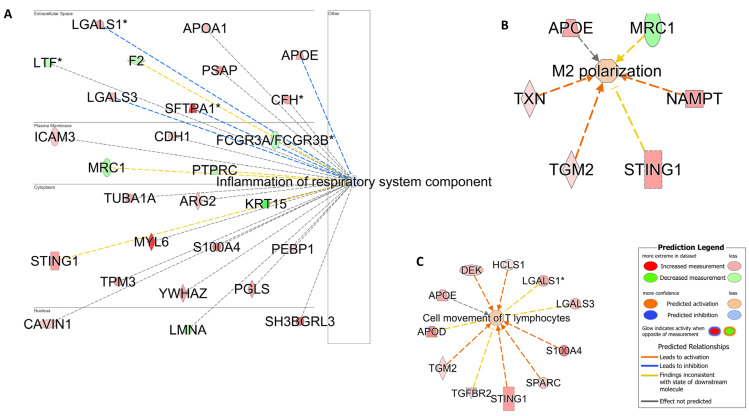
Alternative inflammation profile. Pathway analysis is used to identify networks indicating the inhibition of airway inflammation (**A**), the activation of alternative macrophage polarization (M2) (**B**), and T cell movement (**C**). The biological processes and regulators are colored by their predicted activation state: activated (orange) or inhibited (blue). The darker colors indicate higher scores. The edges connecting the nodes are colored orange when leading to the activation of the downstream node, blue when leading to its inhibition, and yellow if the findings underlying the relationship are inconsistent with the state of the downstream node. The pointed arrowheads indicate that the downstream node is expected to be activated if the upstream node connected to it is activated, whereas the blunt arrowheads indicate that the downstream node is expected to be inhibited if the upstream node that connects to it is activated. The molecules in green are downregulated, and those in red are upregulated. The asterisk (*) indicates that multiple identifiers map to the molecule. The analysis was performed using the Ingenuity Pathway Analysis software v24.0.1.

**Table 1 cells-13-01926-t001:** Groupings based on BALF cytology thresholds.

Group	Number of Animals	Criteria for Inclusion
Group_A_Control	*n* = 8	Neut ≤ 5%; Mast ≤ 2%; Eos ≤ 1%
Group_B_NEUT	*n* = 4	**Neut > 5%**; Mast ≤ 2%; Eos ≤ 1%
Group_C_MAST	*n* = 8	Neut ≤ 5%; **Mast > 2%**; Eos ≤ 1%
Group_D_NEUT_MAST	*n* = 4	**Neut > 5%; Mast > 2%**; Eos ≤ 1%
Group_E_EOS_MAST	*n* = 3	Neut ≤ 5%; **Mast > 2%**; **Eos > 1%**

Neut: neutrophils; Mast: mast cells; Eos: eosinophils.

## Data Availability

The raw RNA-seq data were deposited in the Gene Expression Omnibus under the study accession number GSE277308. The following secure token has been created to allow for the review of record GSE277308 while it remains in private status: mxevuicmjpifdad.

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
