# Peer review of "Distinct Molecular Profiles Underpin Mild-To-Moderate Equine Asthma Cytological Profiles"

_cells, 2024, doi:10.3390/cells13221926_

Round 1
Reviewer 1 Report
Comments and Suggestions for Authors
Mild to moderate equine asthma is a serious problem that is very widespread. However, some aspects of its etiopathogenesis and related prevention are still not fully explained and remain unknown. Cytological profiles in MMEA are quite heterogeneous. The similarity with endotypes of human asthma is assumed, but no study has thoroughly studied and confirmed this. The submitted manuscript fills this fundamental gap in knowledge. Therefore, its great importance and contribution is undoubted. The manuscript significantly helpes to improve the knowledge about the etiopathogenesis of MMEA. Moreover, this is also the first necessary step towards the understanding prevention of MMEA.
BALF samples from 27 horses were analyzed, for which the authors provided all necessary data. The number of samples can be considered relevant. The methodology is described factually and comprehensibly and includes all details. The results are presented in a clear and detailed manner.
The study compared BALF cell gene and protein expression in horses without inflammation with the BALF samples obtained from horses with neutrophilic, mastocytic, mixed neutrophilic/mastocytic and mixed eosinophilic/mastocytic inflammation. Endotype-specific inflammatory pathways were described. The results confirmed the similarity with human asthma endotypes (“Type 2” and “non-Type 2” inflammatory type). This information was unknown and I assume that it will be used in a number of follow-up studies.
Author Response
Comments:
Mild to moderate equine asthma is a serious problem that is very widespread. However, some aspects of its etiopathogenesis and related prevention are still not fully explained and remain unknown. Cytological profiles in MMEA are quite heterogeneous. The similarity with endotypes of human asthma is assumed, but no study has thoroughly studied and confirmed this. The submitted manuscript fills this fundamental gap in knowledge. Therefore, its great importance and contribution is undoubted. The manuscript significantly helpes to improve the knowledge about the etiopathogenesis of MMEA. Moreover, this is also the first necessary step towards the understanding prevention of MMEA.
BALF samples from 27 horses were analyzed, for which the authors provided all necessary data. The number of samples can be considered relevant. The methodology is described factually and comprehensibly and includes all details. The results are presented in a clear and detailed manner.
The study compared BALF cell gene and protein expression in horses without inflammation with the BALF samples obtained from horses with neutrophilic, mastocytic, mixed neutrophilic/mastocytic and mixed eosinophilic/mastocytic inflammation. Endotype-specific inflammatory pathways were described. The results confirmed the similarity with human asthma endotypes (“Type 2” and “non-Type 2” inflammatory type). This information was unknown and I assume that it will be used in a number of follow-up studies.
Response:
The authors are grateful for the reviewer’s recognition of the study's importance in addressing key gaps in the understanding of mild to moderate equine asthma (MMEA) and its comparison with human asthma endotypes.
We appreciate the positive remarks on the relevance of our sample size, the clarity of our methodology, and the presentation of our results. The reviewer’s acknowledgment of the study's contribution to elucidating endotype-specific inflammatory pathways and its potential to inspire follow-up research is particularly encouraging.
Our aim has always been to advance the understanding of the aetiopathogenesis of MMEA and lay the groundwork for future prevention strategies. Feedback like this reassures us of the significance of this work and motivates us to continue contributing to this field.
Thank you once again for the valuable insights and kind words.
Reviewer 2 Report
Comments and Suggestions for Authors
The authors describe transcriptional and proteomic changes in the BALF from horses with moderate to mild horse asthma. They have further divided the animals to 4 different group based on the cellular content in the BALF samples that may reflect different subtypes of the disease. They have performed RNA-seq and mass spectrometry to get gene and protein expression levels and compared each of the group to the control group. Differentially expressed genes (DEGs) were then subjected to gene set enrichment analysis. Two DEGs per comparisons were then validated by quantitative PCR.
The article is written in good English, however, the presentation of the data could be improved to change this very descriptive and redundant structure of describing each comparison separately.
Please, find my comments below.
#1 As mentioned, I would suggest to perform a single analysis (that will also improve FDR values, as for each 4 comparisons, the genes tested should be summed and this sum should be used for multiple testing correction...) where a multifactorial desing is used (please, see DESeq2 vignette for details. This will allow to detect group-specific DEGs. It would also help to structure the manuscript better, with first part describing MMEA effect, and second group-specific effects. Not 4 times group vs control... It is really hard to get to an end reading parts structured exactly the same four times in the row...
#2 Researchers are always interesting in the overlap between DEGs and differentially expressed proteins (DEPs) so I would suggest to create a venn diagram
#3 I do not understand why the RNA-seq reads were subjected to trimming and adaptor removal - with Cutadapt and later trimmomatic - did Cutadapt not performed as expected?
#4 The authors write "genes were detected" - what was the threshold to detect a gene?
#5 Please, distinguish FDR from p-value - I suggest to use "FDR" when referring to adjusted p-value, also in figures when applied
#6 The qPCR results have not been mentioned in the main text. The plot B should start at 0 0 . It would be great to show individual data points as e.g. additional graph.
#7 For the validation with qPCR the data for all 4 groups + control should be shown to validate if a gene is specific to the group.
Minor remarks:
- I suggest to use "RNA-seq" intead of "RNAseq"
- Please, rephrase "Only unique reads" - do the authors mean "uniquely mapped reads"?
- What do authors mean by "differential Cell count"? Should not it be just "Cell count"? Was a statistical test used to test differences here or just a cutoff was set?
- Please consider rephrasing "splice aligner" to "splice-aware aligner"
- Please, use the title case everywhere in the figures or nowhere.
- Please, rephrase "specific-strand"
- What do the authors refer to as saying "35M coverage" in case of transcriptome?
- Please, consider "log2" instead of "Log2" and please make sure to indicate, where necessary, the absolute value and correct the axis labels in supplementary material
- Please, make sure the gene symbols are in italics when refer to a gene
Author Response
The authors would like to thank you very much for giving us the opportunity to submit a revised draft of our manuscript titled
“Distinct molecular profiles underpin mild to moderate equine asthma cytological profiles”
We are grateful to the reviewers’ comments and recommendations and have incorporated changes in the manuscript to reflect the suggestions provided by the reviewers. Here is a point-by-point response to the reviewers’ comments and concerns. We have highlighted the recommended changes within the manuscript using tracked changes.
The authors describe transcriptional and proteomic changes in the BALF from horses with moderate to mild horse asthma. They have further divided the animals to 4 different group based on the cellular content in the BALF samples that may reflect different subtypes of the disease. They have performed RNA-seq and mass spectrometry to get gene and protein expression levels and compared each of the group to the control group. Differentially expressed genes (DEGs) were then subjected to gene set enrichment analysis. Two DEGs per comparisons were then validated by quantitative PCR.
The article is written in good English, however, the presentation of the data could be improved to change this very descriptive and redundant structure of describing each comparison separately.
Please, find my comments below.
#1 As mentioned, I would suggest to perform a single analysis (that will also improve FDR values, as for each 4 comparisons, the genes tested should be summed and this sum should be used for multiple testing correction...) where a multifactorial desing is used (please, see DESeq2 vignette for details. This will allow to detect group-specific DEGs. It would also help to structure the manuscript better, with first part describing MMEA effect, and second group-specific effects. Not 4 times group vs control... It is really hard to get to an end reading parts structured exactly the same four times in the row...
Response: The authors would like to thank the reviewer for his thoughtful and detailed review of our manuscript.
Regarding the reviewers' comment on performing a single-run DESeq2 analysis, we appreciate the potential advantages of such an approach in terms of multiple testing correction. As described in the “Processing of RNA sequencing data and differential expression analysis” section in the Materials and Methods, we applied DESeq2, to perform a comparison of gene expression between the groups of samples. A PCA analysis was performed using the "plotPCA" function within the DESeq2 R package [1]. The Wald test was used to generate p-values and log2 fold changes. Genes with adjusted p-values or false discovery rate (FDR) ≤ 0.05 were identified as differentially expressed genes for each comparison. FDR is crucial for reducing the risk of false discoveries while maintaining the ability to detect meaningful changes in RNA-seq experiments. Therefore, multiple test correction has been applied in all comparisons performed. Moreover, the validity of the data, in terms of differential expression, was further supported by the qPCR results, where eight genes were tested in total across all comparisons.
Our bioinformaticians at Genewiz (now Azenta; https://www.azenta.com/), experts in RNA and single-cell sequencing and data analysis, advised this method for our study. Since the samples were not derived from a paired or longitudinal study, applying a more stringent single-run analysis would likely result in the omission of important information, leading to a more limited list of differentially expressed genes which we felt would potentially detract from the principal aim of the study; namely to reveal MMEA “type-specific” molecular pathways and signatures.
Additionally, we sought further advice from Edinburgh Genomics (https://genomics.ed.ac.uk/) at the University of Edinburgh, with whom we are collaborating on a spatial transcriptomics project. They provided similar recommendations, reinforcing our decision to proceed with our current analytical approach.
It is also worth noting that the results presented in this manuscript could have been presented in the form of four separate manuscripts, each focusing on distinct aspects of the analysis. However, we chose to combine them to provide a more comprehensive understanding of the different MMEA “types”, aligning with the broader aim of the study.
We did perform the analysis of all groups versus the control, as recommended by the reviewer, to examine the MMEA effect. However, this analysis yielded a limited number of differentially expressed genes, likely due to the high variability between individuals and the distinct disease endotypes as well as the likely inevitable overlap between “types”. As such, we decided not to include this data in the manuscript, as we believe it would contribute little to the overall findings and could potentially introduce confusion.
It is important to note that this was not the primary aim of the current study. Our objective is to describe the distinct molecular profiles associated with the different MMEA BALF cytological profiles, which currently constitutes the principal (and only) limited means by which MMEA subtypes are recognised in a clinical setting. We aimed to interrogate this cytological variation further with a view to revealing potential aetiological candidates and therapeutic targets; we believe this provides greater relevance and value to the study's purpose.
Finally, regarding the structure of the Results section, we apologise for any confusion caused. We understand the challenges associated with presenting such large-scale multi-omic data. To address this, we adhered to the logical flow principle commonly recommended in scientific writing and we ensured consistency by using the same structure/order and subsections across all comparisons (e.g., RNA-seq, proteomics, and pathway analyses) [2].
#2 Researchers are always interesting in the overlap between DEGs and differentially expressed proteins (DEPs) so I would suggest to create a venn diagram
Response: Thank you for your comment. We have performed the Venn diagrams between transcriptomic and proteomic data of the different groups and, in agreement with our findings from previously published studies, the degree of overlap was negligible. This observation is not unexpected as such a lack of concordance between gene and protein expression has been well documented by others [3, 4] and our group [5] in similar datasets and can be attributed to a variety of factors such as post-transcriptional machinery, variable half-lives, molecular degradation, or even sampling bias [6]. Certain factors specific to the current study design may also have contributed to this discordance in gene and protein expression. Firstly, there was a remarkably limited number of DEGs (range 18 – 34) versus the DEPs (217 - 359) detected in this study. Secondly, unlike the transcriptomic analysis, proteomic analysis was not restricted to the cellular component of the BALF sample, thus permitting the detection of secreted proteins. This approach was considered more likely to provide a more holistic assessment of the airway immune status as well as offering a greater potential to identify biomarkers with potential clinical and/or training applications. Moreover, with such multi-omic studies, the conclusions that can be derived from the data generated are highly dependent on the quality of the annotations of the genome used, which for the horse remains challenging [7]. For the above reasons, we opted not to include Venn diagrams depicting the level of overlap. Rather, we considered the lack of concordance to highlight the value of applying a combined transcriptomic and proteomic approach when studying cellular mechanisms with a view to unravelling disease pathogeneses.
#3 I do not understand why the RNA-seq reads were subjected to trimming and adaptor removal - with Cutadapt and later trimmomatic - did Cutadapt not performed as expected?
Response: Thank you very much for your comment and apologies for the confusion. Initially, we used Cutadapt for data cleaning, but later switched to Trimmomatic on raw sequencing reads based on recommendation from Genewiz. This has now been amended, to avoid confusion.
#4 The authors write "genes were detected" - what was the threshold to detect a gene?
Response: Thank you for your question. Adjusted p-value ≤ 0.05 was identified as differentially expressed genes for each comparison. This is stated in the section “Processing of RNA sequencing data and differential expression analysis” in the Materials and Methods.
#5 Please, distinguish FDR from p-value - I suggest to use "FDR" when referring to adjusted p-value, also in figures when applied
Response: Thank you for your recommendation. We agree with the reviewer. FDR refers to adjusted p-value throughout the manuscript and Figure legends have now been amended.
#6 The qPCR results have not been mentioned in the main text. The plot B should start at 0 0 . It would be great to show individual data points as e.g. additional graph.
Response: Thank you for your comment. Plot B is starting at 0, now number 0 has been added on the axis to avoid confusion. qPCR results are mentioned in the section “3.2 Total RNA and protein extraction of bronchoalveolar lavage samples has been successfully performed”, as well as in the Discussion section (lines 573-575, 627-629). Given the large number of figures already included in the manuscript (10 in total) and the decreasing reliance on qPCR validation (please see comment below)[8], we have decided to present the qPCR data as supplementary material. Individual data points are displayed in Plot B, and an additional graph presenting the qPCR data separately has been included in the Supplementary material 1, as recommended by the Reviewer 2.
#7 For the validation with qPCR the data for all 4 groups + control should be shown to validate if a gene is specific to the group.
Response: qPCR validation data for all four groups versus control are shown in supplementary material 1. The data are consistent with the results of RNA-seq analysis for these genes in the same samples; IDO1, THBS1 from Group_B_NEUT versus Control, KIT, ROS1 from Group_C_MAST versus Control, CCL11, TREM1 from Group_D_NEUT_MAST versus Control, and CD28, LPO from Group_E_EOS_MAST versus Control. RNA-seq methods and the associated data analysis are now widely regarded as robust enough to not always require validation by qPCR or other approaches [8]. Nonetheless, we performed gene validation using qPCR for a subset of genes across all comparisons.
Minor remarks:
- I suggest to use "RNA-seq" intead of "RNAseq"
Response: Thank you for your suggestion. This has now been amended accordingly through the manuscript and supplementary data.
- Please, rephrase "Only unique reads" - do the authors mean "uniquely mapped reads"?
Response: Correct, this has now been amended.
- What do authors mean by "differential Cell count"? Should not it be just "Cell count"? Was a statistical test used to test differences here or just a cutoff was set?
Response: The differential cell count of bronchoalveolar lavage fluid (BALF) is a well-established term in equine and human medicine that has been routinely used for decades [9, 10]. It reflects the “relative” cellular composition of the lower respiratory tract and provides valuable insights into respiratory health. The differential cell count of BALF is a valuable diagnostic tool, utilising cut-off values to identify respiratory conditions and guide treatment plans. In the case of equine asthma at present it consists the gold standard diagnostic criterion along with history and clinical signs [10].
Cut-off values were applied to group the samples, as detailed in the “Horses used in this study” and “Sample Collection” sections of the Materials and Methods. Mild to moderate equine asthma was defined according to the American College of Veterinary Internal Medicine guidelines by the following BALF cell ratios in horses without systemic signs of disease or increased respiratory effort at rest: neutrophils > 5% and/or mast cells > 2% and/or eosinophils >1% [11-15].
- Please consider rephrasing "splice aligner" to "splice-aware aligner"
Response: This has now been rephrased. Thank you.
- Please, use the title case everywhere in the figures or nowhere.
Response: All figure legends follow the same format. Apologies for the confusion.
- Please, rephrase "specific-strand"
Response: This has now been rephrased to “strand-specific” [16], as requested.
- What do the authors refer to as saying "35M coverage" in case of transcriptome?
Response: This indicates that the sequencing run generated 35 million (35M) individual sequencing reads. It refers to the sequencing depth or the number of reads (35 million reads) generated through the sequencing process. It is a specific attribute of sequencing data routinely used to describe high-throughput RNA-seq experiments, where 35 million reads are generally recommended. This has now been indicated in the text (line 327). Thank you.
- Please, consider "log2" instead of "Log2" and please make sure to indicate, where necessary, the absolute value and correct the axis labels in supplementary material
Response: These have now been amended and the axis labels have been corrected in the supplementary data, as well, as requested by the reviewer. Thank you.
- Please, make sure the gene symbols are in italics when refer to a gene
Response: Thank you for your comment. That is applied throughout the manuscript.
The authors would like to thank you once again for your comments.
We look forward to hearing from you in due time regarding our submission and to respond to any further questions and comments you may have.
Sincerely,
Anna Eleonora Karagianni
DVM, MRes, PhD, MRCVS, AFHEA
Lecturer in Veterinary Clinical Research
Department of Veterinary Clinical Sciences
School of Veterinary Medicine
Faculty of Health and Medical Sciences
University of Surrey, UK
References
- Love MI, Huber W, Anders S: Moderated estimation of fold change and dispersion for RNA-seq data with DESeq2. Genome Biol 2014, 15(12):550.
- Barroga E, Matanguihan GJ: Creating Logical Flow When Writing Scientific Articles. J Korean Med Sci 2021, 36(40):e275.
- Chen G, Gharib TG, Huang CC, Taylor JM, Misek DE, Kardia SL, Giordano TJ, Iannettoni MD, Orringer MB, Hanash SM, Beer DG: Discordant protein and mRNA expression in lung adenocarcinomas. Mol Cell Proteomics 2002, 1(4):304-313.
- Ghazalpour A, Bennett B, Petyuk VA, Orozco L, Hagopian R, Mungrue IN, Farber CR, Sinsheimer J, Kang HM, Furlotte N et al: Comparative Analysis of Proteome and Transcriptome Variation in Mouse. PLOS Genetics 2011, 7(6):e1001393.
- Karagianni AE, Kurian D, Cillán-Garcia E, Eaton SL, Wishart TM, Pirie RS: Training associated alterations in equine respiratory immunity using a multiomics comparative approach. Sci Rep 2022, 12(1):427.
- Haider S, Pal R: Integrated Analysis of Transcriptomic and Proteomic Data. Current Genomics 2013, 14(2):91-110.
- Kalbfleisch TS, Rice ES, DePriest MS, Jr., Walenz BP, Hestand MS, Vermeesch JR, BL OC, Fiddes IT, Vershinina AO, Saremi NF et al: Improved reference genome for the domestic horse increases assembly contiguity and composition. Commun Biol 2018, 1:197.
- Coenye T: Do results obtained with RNA-sequencing require independent verification? Biofilm 2021, 3:100043.
- De Brauwer EI, Drent M, Mulder PG, Bruggeman CA, Wagenaar SS, Jacobs JA: Differential cell analysis of cytocentrifuged bronchoalveolar fluid samples affected by the area counted. Anal Quant Cytol Histol 2000, 22(2):143-149.
- Simões J, Batista M, Tilley P: The Immune Mechanisms of Severe Equine Asthma-Current Understanding and What Is Missing. Animals (Basel) 2022, 12(6).
- Bedenice D, Mazan MR, Hoffman AM: Association between cough and cytology of bronchoalveolar lavage fluid and pulmonary function in horses diagnosed with inflammatory airway disease. Journal Of Veterinary Internal Medicine / American College Of Veterinary Internal Medicine 2008, 22(4):1022-1028.
- Lavoie JP, Cesarini C, Lavoie-Lamoureux A, Moran K, Lutz S, Picandet V, Jean D, Marcoux M: Bronchoalveolar Lavage Fluid Cytology and Cytokine Messenger Ribonucleic Acid Expression of Racehorses with Exercise Intolerance and Lower Airway Inflammation. Journal of Veterinary Internal Medicine 2011, 25(2):322-329.
- Davis KU, Sheats MK: Differential gene expression and Ingenuity Pathway Analysis of bronchoalveolar lavage cells from horses with mild/moderate neutrophilic or mastocytic inflammation on BAL cytology. Vet Immunol Immunopathol 2021, 234:110195.
- Couetil LL, Cardwell JM, Gerber V, Lavoie JP, Leguillette R, Richard EA: Inflammatory Airway Disease of Horses--Revised Consensus Statement. 2016, 30(2):503-515.
- Couëtil LL, Cardwell JM, Gerber V, Lavoie JP, Léguillette R, Richard EA: Inflammatory Airway Disease of Horses--Revised Consensus Statement. J Vet Intern Med 2016, 30(2):503-515.
- Borodina T, Adjaye J, Sultan M: A strand-specific library preparation protocol for RNA sequencing. Methods Enzymol 2011, 500:79-98.